# Multimodal Variational Autoencoders for Semi-Supervised Learning: In Defense of Product-of-Experts

## Abstract

Multimodal generative models should be able to learn a meaningful latent representation that enables a coherent joint generation of all modalities (e.g., images and text). Many applications also require the ability to accurately sample modalities conditioned on observations of a subset of the modalities. Often not all modalities may be observed for all training data points, so semi-supervised learning should be possible. In this study, we evaluate a family of product-of-experts (PoE) based variational autoencoders that have these desired properties. We include a novel PoE based architecture and training procedure. An empirical evaluation shows that the PoE based models can outperform an additive mixture-of-experts (MoE) approach. Our experiments support the intuition that PoE models are more suited for a conjunctive combination of modalities while MoEs are more suited for a disjunctive fusion.

## 1 Introduction

Multimodal generative modelling is important because information about real-world objects typically comes in different representations, or modalities. The information provided by each modality may be erroneous and/or incomplete, and a complete reconstruction of the full information can often only be achieved by combining several modalities. For example, in image- and video-guided translation (Caglayan et al., 2019), additional visual context can potentially resolve ambiguities (e.g., noun genders) when translating written text.

In many applications, modalities may be missing for a subset of the observed samples during training and deployment. Often the description of an object in one modality is easy to obtain, while annotating it with another modality is slow and expensive. Given two modalities, we call samples *paired* when both modalities are present, and *unpaired* if one is missing. The simplest way to deal with paired and unpaired training examples is to discard the unpaired observations for learning. The smaller the share of paired samples, the more important becomes the ability to additionally learn from the unpaired data, referred to as *semi-supervised learning* in this context (following the terminology from Wu & Goodman, 2019. Typically one would associate semi-supervised learning with learning form labelled and unlabelled data to solve a classification or regression tasks). Our goal is to provide a model that can leverage the information contained in unpaired samples and to investigate the capabilities of the model in situations of low levels of supervision, that is, when only a few paired samples are available. While a modality can be as low dimensional as a label, which can be handled by a variety of discriminative models (van Engelen & Hoos, 2020), we are interested in high dimensional modalities, for example an image and a text caption.

Learning a representation of multimodal data that allows to generate high-quality samples requires the following: 1) deriving meaningful representation in a joint latent space for each high dimensional modality and 2) bridging the representations of different modalities in a way that the relations between them are preserved. The latter means that we do not want the modalities to be represented orthogonally in the latent space – ideally the latent space should encode the object's properties independent of the input modality. Variational autoencoders (Kingma & Welling, 2014) using a product-of-experts (PoE, Hinton, 2002; Welling, 2007) approach for combining input modalities are a promising approach for multimodal generative modelling having the desired properties, in partic-

ular the VAEVAE model developed by Wu & Goodman (2018) and a novel model termed SVAE, which we present in this study. Both models can handle multiple high dimensional modalities, which may not all be observed at training time.

It has been argued that a product-of-experts (PoE) approach is not well suited for multimodal generative modelling using variational autoencoders (VAEs) in comparison to additive mixture-of-experts (MoE). It has empirically been shown that the PoE-based MVAE (Wu & Goodman, 2018) fails to properly model two high-dimensional modalities in contrast to an (additive) MoE approach referred to as MMVAE, leading to the conclusion that "PoE factorisation does not appear to be practically suited for multi-modal learning" (Shi et al., 2019). This study sets out to test this conjecture for state-of-the-art multimodal VAEs.

The next section summarizes related work. Section 3 introduces SVAE as an alternative PoE based VAE approach derived from axiomatic principles. Then we present our experimental evaluation of multimodal VAEs before we conclude.

## 2 Background and Related Work

We consider multimodal generative modelling. We mainly restrict our considerations to two modalities $x_1 \in X_1, x_2 \in X_2$, where one modality may be missing at a time. Extensions to more modalities are discussed in Experiments section and Appendix D. To address the problem of generative cross-modal modeling, one modality $x_1$ can be generated from another modality $x_2$ by simply using independently trained generative models ($x_1 \rightarrow x_2$ and $x_2 \rightarrow x_1$) or a composed but non-interchangeable representation (Wang et al., 2016; Sohn et al., 2015). However, the ultimate goal of multimodal representation learning is to find a meaningful joint latent code distribution bridging the two individual embeddings learned from $x_1$ and $x_2$ alone. This can be done by a two-step procedure that models the individual representations first and then applies an additional learning step to link them (Tian & Engel, 2019; Silberer & Lapata, 2014; Ngiam et al., 2011). In contrast, we focus on approaches that learn individual and joint representations simultaneously. Furthermore, our model should be able to learn in a semi-supervised setting. Kingma et al. (2014) introduced two models suitable for the case when one modality is high dimensional (e.g., an image) and another is low dimensional (e.g., a label) while our main interest are modalities of high complexity.

We consider models based on variational autoencoders (VAEs, Kingma & Welling, 2014; Rezende et al., 2014). Standard VAEs learn a latent representation $z \in Z$ for a set of observed variables $x \in X$ by modelling a joint distribution $p(x, z) = p(z)p(x|z)$. In the original VAE, the intractable posterior $q(z|x)$ and conditional distribution $p(x|z)$ are approximated by neural networks trained by maximising the ELBO loss taking the form

$$\mathcal{L} = E_{q(z|x)}[\log p(x|z)] - D_{\text{KL}}(q(z|x) \parallel \mathcal{N}(0, I)) \qquad (1)$$

with respect to the parameters of the networks modelling $q(z|x)$ and $p(x|z)$. Here $D_{\text{KL}}(\cdot \parallel \cdot)$ denotes the Kullback-Leibler divergence. Bi-modal VAEs that can handle a missing modality extend this approach by modelling $q(z|x_1, x_2)$ as well as $q_1(z|x_1)$ and $q_2(z|x_2)$, which replace the single $q(z|x)$. Multimodal VAEs may differ in 1) the way they approximate $q(z|x_1, x_2)$, $q_1(z|x_1)$ and $q_2(z|x_2)$ by neural networks and/or 2) the structure of the loss function, see Figure 1. Typically, there are no conceptual differences in the decoding, and we model the decoding distributions in the same way for all methods considered in this study.

Suzuki et al. (2017) introduced a model termed JMVAE (Joint Multimodal VAE), which belongs to the class of approaches that can only learn from the paired training samples (what we refer to as the *(fully) supervised setting*). It approximates $q(z|x_1, x_2)$, $q_1(z|x_1)$ and $q_2(z|x_2)$ with three corresponding neural networks and optimizes an ELBO-type loss of the form

$$\mathcal{L} = E_{q(z|x_1,x_2)}[\log p_1(x_1|z) + \log p_2(x_2|z)] - D_{\text{KL}}(q(z|x_1, x_2) \parallel \mathcal{N}(0, I))$$
$$- D_{\text{KL}}(q(z|x_1, x_2) \parallel q_1(z|x_1)) - D_{\text{KL}}(q(z|x_1, x_2) \parallel q_2(z|x_2)) \ . \quad (2)$$

The last two terms imply that during learning the joint network output must be generated which requires paired samples.

The MVAE (Multimodal VAE) model (Wu & Goodman, 2018) is the first multimodal VAE-based model allowing for missing modalities that does not require any additional network structures for

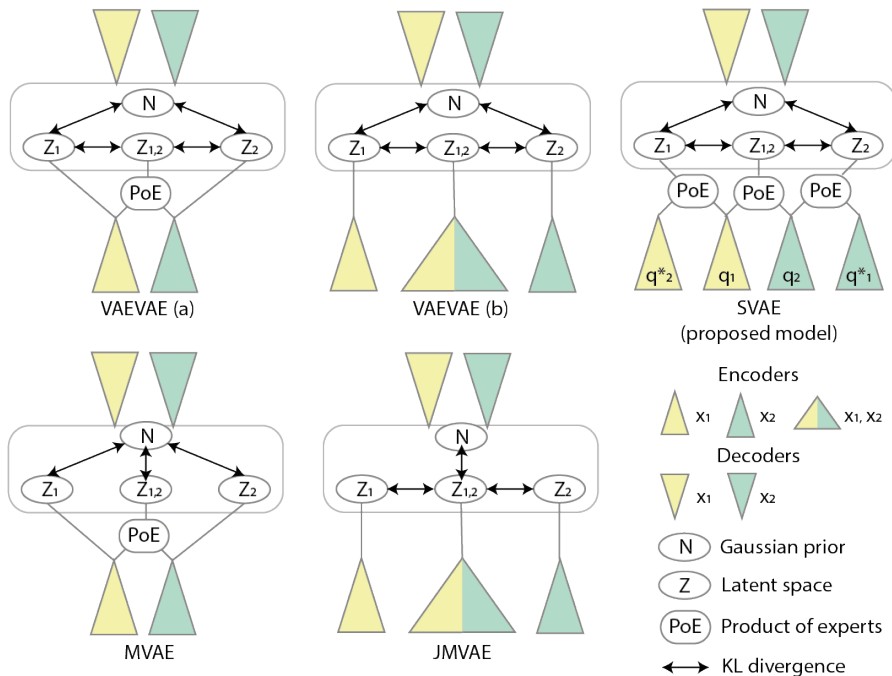

Figure 1: Schematic overview bi-modal VAEs using a PoE and additional network structures that are capable of semi-supervised learning without requiring a two step learning procedure. VAEVAE (a) and (b) are by Wu & Goodman (2019), JMVAE is by Suzuki et al. (2017), MVAE is by Wu & Goodman (2018), and SVAE is our newly proposed model. Each triangle stands for an individual neural network, the colors indicate the two different modalities.

learning the joint latent code distribution. The joint posterior is modeled using a product-of-experts (PoE) as $q(z|x_{1:M}) = \prod_m q_m(z|x_m)$. For the missing modality $q_k(z|x_k) = 1$ is assumed. The model allows for semi-supervised learning while keeping the number of model parameters low.

The bridged model (Yadav et al., 2020) highlights the need for an additional network structure for approximating the joint latent code distribution. It attempts to keep the advantages of the additional encoding networks. It reduces the number of model parameters by introducing the *bridge encoder* that consists of one fully connected layer which takes $z_1$ and $z_2$ latent code vectors generated from $x_1$ and $x_2$ and outputs the mean and the variance of the joint latent code distribution.

The arguably most advanced multimodal VAE models is VAEVAE by Wu & Goodman (2019), which we discuss in detail in the next section (see also Appendix C).

Shi et al. (2019) proposed a MoE model termed MMVAE (Mixture-of-experts Multimodal VAE). In MMVAE model the joint variational posterior for $M$ modalities is approximated as $q(z|x_{1:M}) = \sum_m \alpha_m q_m(z|x_m)$ where $\alpha_m = \frac{1}{M}$. The model utilizes a loss function from the importance weighted autoencoder (IWAE, Burda et al., 2016) that computes a tighter lower bound compared to the VAE ELBO loss. The MoE rule formulation allows in principle to train with a missing modality $i$ by assuming $\alpha_i = 0$, however, Shi et al. (2019) do not highlight or evaluate this feature. There are benchmarks in the paper that compare MVAE (Wu & Goodman, 2018) and MMVAE, concluding that MVAE often fails to learn the joint latent code distribution. Because of these results and those presented by Wu & Goodman (2019), we did not include MVAE as a benchmark model in our experiments.

## 3 VAEVAE AND SVAE

We developed a new approach as an alternative to VAEVAE. Both models 1) are VAE based; 2) allow for interchangeable cross-model generation as well as a learning joint embedding; 3) allow for missing modalities at training time; and 4) can be applied to two similarly complex high dimensional

modalities. Next, we will briefly present our new model SVAE. Then we highlight the differences to VAEVAE. Finally, we state a newly derived objective function for training the models. We consider two modalities and refer to Appendix D for generalizations to more modalities.

**SVAE**  Since both modalities might not be available for all the samples, it should be possible to marginalize each of them out of $q(z|x_1, x_2)$. While the individual encoding distributions $q(z|x_1)$ and $q(z|x_2)$ can be approximated by neural networks as in the standard VAE, we need to define a meaningful approximation of the joint encoding distribution $q(z|x) = q(z|x_1, x_2)$. In the newly proposed SVAE model, these distributions are defined as the following:

$$q(z|x_1, x_2) = \frac{1}{Z(x_1, x_2)} q_1(z|x_1) q_2(z|x_2) \tag{3}$$

$$q(z|x_1) = q_1(z|x_1) q_2^*(z|x_1) \tag{4}$$

$$q(z|x_2) = q_2(z|x_2) q_1^*(z|x_2) \tag{5}$$

$$q(z) = \mathcal{N}(0, I) \tag{6}$$

The model is derived from an axiomatic proof that is given in Appendix A. The desired properties of the model were that 1) when no modalities are observed the generating distribution for the latent code is Gaussian, 2) the modalities are independent given the latent code, 3) both experts cover the whole latent space with equal probabilities, and 4) the joint encoding distribution $q(z|x_1, x_2)$ is modelled by a PoE.

The distributions $q_1(z|x_1)$, $q_2(z|x_2)$, $q_2^*(z|x_1)$ and $q_1^*(z|x_2)$ are approximated by neural networks. In case both observations are available, $q(z|x_1, x_2)$ is approximated by applying the product-of-experts rule with $q_1(z|x_1)$ and $q_2(z|x_2)$ being the experts for each modality. In case of a missing modality, equation 4 or 5 is used. If, for example, $x_2$ is missing, the $q_2^*(z|x_1)$ distribution takes over as a "replacement" expert, modelling marginalization over $x_2$.

**SVAE vs. VAEVAE**  The VAEVAE model (Wu & Goodman, 2019) is the most similar to ours. Wu & Goodman define two variants which can be derived from the SVAE model in the following way. Variant (a) can be derived by setting $q^*(z|x_1) = q^*(z|x_2) = 1$. Variant (b) is obtained from (a) by additionally using a separate network to model $q(z|x_1, x_2)$. Having a joint network $q(z|x_1, x_2)$ implements the most straightforward way of capturing the inter-dependencies of the two modalities. However, the joint network cannot be trained on unpaired data – which can be relevant when the share of supervised data gets smaller. Option (a) uses the product-of-experts rule to model the joint distribution of the two modalities as well, but does not ensure that both experts cover the whole latent space (in contrast to SVAE, see equation A.14 in the appendix), which can lead to individual latent code distributions diverging. Based on this consideration and the experimental results from Wu & Goodman (2019), we focused on benchmarking VAEVAE (b) and refer to it as simply VAEVAE in Section 4.

SVAE resembles VAEVAE in the need for additional networks besides one encoder per each modality and the structure of ELBO loss. It does, however, solve the problem of learning the joint embeddings in a way that allows to learn the parameters of approximated $q(z|x_1, x_2)$ using all available samples, i.e., both paired and unpaired. If $q(z|x_1, x_2)$ is approximated with the joint network that accepts concatenated inputs, as in JMVAE and VAEVAE (b), the weights of $q(z|x_1, x_2)$ can only be updated for the paired share of samples. If $q(z|x_1, x_2)$ is approximated with a PoE of decoupled networks as in SVAE, the weights are updated for each sample whether paired or unpaired – which is the key differentiating feature of SVAE compared to existing architectures.

**A New Objective Function**    When developing SVAE, we devised a novel ELBO-type loss:

$$
\begin{aligned}
\mathcal{L} =\ & E_{p_{\text{paired}}(x_1,x_2)} \big[ E_{q(z|x_1,x_2)}[\log p_1(x_1|z) + \log p_2(x_2|z)] \big] \\
& - D_{\text{KL}}(q(z|x_1,x_2) \parallel p(z|x_1)) - D_{\text{KL}}(q(z|x_1,x_2) \parallel p(z|x_2)) \\
& + E_{p_{\text{paired}}(x_1)} \big[ E_{q(z|x_1)}[\log p_1(x_1|z)] - D_{\text{KL}}(q(z|x_1) \parallel p(z)) \big] \\
& + E_{p_{\text{paired}}(x_2)} \big[ E_{q(z|x_2)}[\log p_2(x_2|z)] - D_{\text{KL}}(q(z|x_2) \parallel p(z)) \big]
\end{aligned}
\tag{7}
$$

$$
\mathcal{L}_1 = E_{p_{\text{unpaired}}(x_1)} \big[ E_{q(z|x_1)}[\log p_1(x_1|z)] - D_{\text{KL}}(q(z|x_1) \parallel p(z)) \big]
\tag{8}
$$

$$
\mathcal{L}_2 = E_{p_{\text{unpaired}}(x_2)} \big[ E_{q(z|x_2)}[\log p_2(x_2|z)] - D_{\text{KL}}(q(z|x_2) \parallel p(z)) \big]
\tag{9}
$$

$$
\mathcal{L}_{\text{comb}} = \mathcal{L} + \mathcal{L}_1 + \mathcal{L}_2
\tag{10}
$$

Here $p_{\text{paired}}$ and $p_{\text{unpaired}}$ denote the distributions of the paired and unpaired training data, respectively. The loss function is derived in Appendix B. The differences between this loss and the loss function used to train VAEVAE by Wu & Goodman (2019) are highlighted in Appendix C.

## 4 EXPERIMENTS

We conducted experiments to compare state-of-the-art PoE based VAEs with the MoE approach MMVAE (Shi et al., 2019). We considered VAEVAE (b) as proposed by Wu & Goodman (2019) and SVAE as described above. The two approaches differ both in the underlying model as well as the objective function. For a better understanding of these differences, we also considered an algorithm referred to as VAEVAE*, which has the same model architecture as VAEVAE and the same loss function as SVAE.[1] The difference in the training procedure for VAEVAE and VAEVAE* is described in Appendix C. Since the VAEVAE implementation was not publicly available at the time of writing, we used our own implementation of VAEVAE based on the PiXYZ library.[2] For details about the experiments we refer to Appendix F. The source code to reproduce the experiments can be found in the supplementary material.

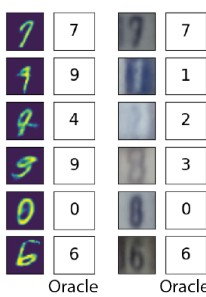

Figure 2: MNIST-SVHN reconstruction for fully supervised VAEVAE.

For an unbiased evaluation, we considered the same test problems and performance metrics as Shi et al. (2019). In addition, we designed an experiment referred to as MNIST-Split that was supposed to be well-suited for PoE. In all experiments we kept the network architectures as similar as possible (see Appendix F). For the new benchmark problem, we constructed a multi-modal dataset where the modalities are similar in dimensionality as well as complexity and are providing missing information to each other rather than duplicating it. The latter should favor a PoE modelling, which suits an "AND" combination of the modalities, and not a MoE modeling, which is more aligned with an "OR" combination.

We measured performance for different supervision levels for each dataset (e.g., 10% supervision level means that 10% of the training set samples were paired and the remaining 90% were unpaired).

**Image and image: MNIST-Split**    We created an image reconstruction dataset based on MNIST digits (LeCun et al., 1998). The images were split horizontally into equal parts, either two or three depending on the experimental setting. These regions are considered as different input modalities. In the above notion of "AND" and "OR" tasks we implicitly assume an additional modality, which is the image label in this case. The fact that the correct digit can sometimes be guessed from only one part of the image makes the new MNIST-Split benchmark a mixture of an "AND" and an "OR" task. This is in contrast to the MNIST-SVHN task described below, which can be regarded as an almost pure "OR" task.

TWO MODALITIES: MNIST-SPLIT.    In the bi-modal version referred to as MNIST-Split, the MNIST images were split in top and bottom halves of equal size, and the halves were then used

---

[1] We also evaluated SVAE*, our model with the VAEVAE loss function, but it never outperformed other models.

[2] https://github.com/masa-su/pixyz

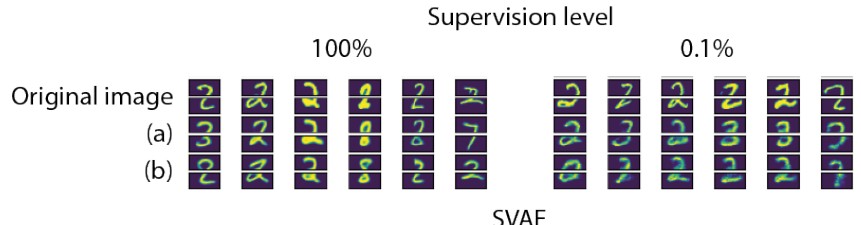

Figure 3: MNIST-Split image reconstructions of a top half and a bottom half given (a) the top half; (b) the bottom half of the original image.

as two modalities. We tested the quality of the image reconstruction given one or both modalities by predicting the reconstructed image label with an independent oracle network, a ResNet-18 (He et al., 2016) trained on the original MNIST dataset. The evaluation metrics were *joint coherence*, *synergy*, and *cross-coherence*. For measuring joint coherence, 1000 latent space vectors were generated from the prior and both halves of an image were then reconstructed with the corresponding decoding networks. The concatenated halves yield the fully reconstructed image. Since the ground truth class labels do not exist for the randomly sampled latent vectors, we could only perform a qualitative evaluation, see Figure 3. Synergy was defined as the accuracy of the image reconstruction given both halves. Cross-coherence considered the reconstruction of the full image from one half and was defined as the fraction of class labels correctly predicted by the oracle network.

Table 1: Evaluation of the models trained on the fully supervised datasets.

|  | Accuracy (both) | Accuracy (top half) | Accuracy (bottom half) |
|---|---|---|---|
| MMVAE | 0.539 | 0.221 | 0.283 |
| SVAE | 0.948 | 0.872 | 0.816 |
| VAEVAE | 0.956 | 0.887 | 0.830 |
| VAEVAE* | 0.958 | 0.863 | 0.778 |

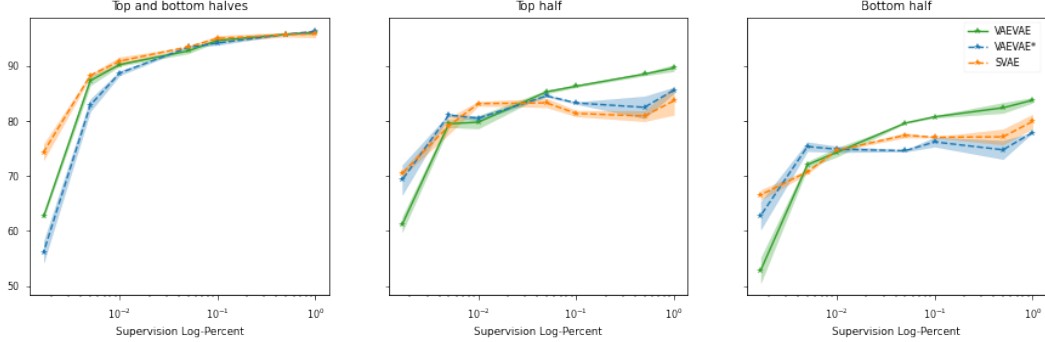

Figure 4: MNIST-Split dataset. Accuracy of an oracle network applied to images reconstructed given (a) the full image (both halves) (b) the top half (c) the bottom half.

The quantitative results are shown in Table 1 and Figure 4. All PoE architectures clearly outperformed MMVAE even when trained on the low supervision levels. In this experiment, it is important that both experts agree on a class label. Thus, as expected, the multiplicative PoE fits the task much better than the additive mixture. Utilizing the novel loss function (10) gave the best results for very low supervision (SVAE and VAEVAE*).

THREE MODALITIES: MNIST-SPLIT-3. We compared a simple generalization of the SVAE model to more than two modalities with the canonical extension of the VAEVAE model (both defined in Appendix D) on the MNIST-Split-3 data, the 3-modal version of MNIST-Split task. Figure 6

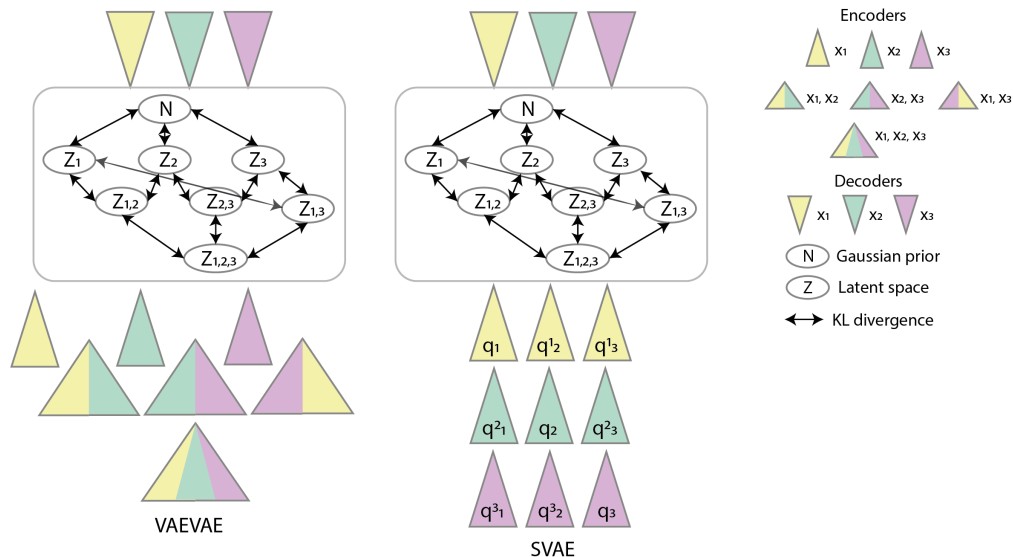

Figure 5: The SVAE and VAEVAE network architectures for 3-modalities case. The number of parameters is $kn^2$ for SVAE and $kn2^{n-1}$ for VAEVAE, where $n$ is the number of modalities and $k$ is the number of parameters in one encoding network.

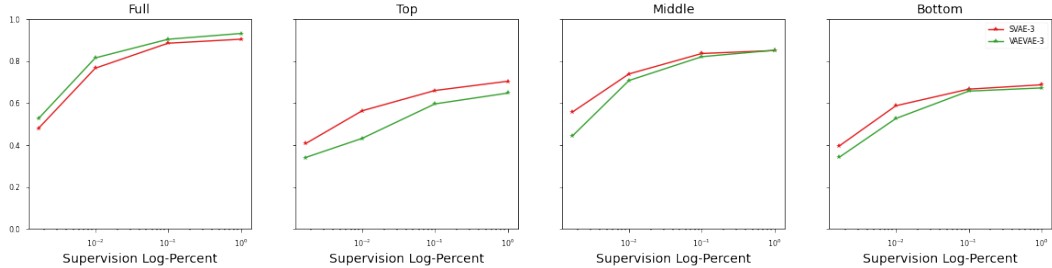

Figure 6: MNIST-Split-3 dataset, reproducing the logic of MNIST-Split for the images splitted in three parts.

shows that SVAE performed better when looking at the individual modalities reconstructions. While the number of parameters in the bi-modal case is the same for SVAE and VAEVAE, it grows exponentially for VAEVAE and stays in order of $n^2$ for SVAE where $n$ is the number of modalities, see Figure 5 and Appendix D for details.

**Image and image: MNIST-SVHN** The first dataset considered by Shi et al. (2019) is constructed by pairing MNIST and SVHN (Netzer et al., 2011) images showing the same digit. This dataset shares some properties with MNIST-Split, but the relation between the two modalities is different: the digit class is derived from a concatenation of two modalities in MNIST-Split, while in MNIST-SVHN it could be derived from any modality alone, which corresponds to "AND" combination of the modalities and favors the MoE architecture. As before, oracle networks are trained to predict the digit classes of MNIST and SVHN images. Joint coherence was again computed based on 1000 latent space vectors generated from the prior. Both images were then reconstructed with the corresponding decoding networks. A reconstruction was considered correct if the predicted digit classes of MNIST and SVHN were the same. Cross-coherence was measured as above.

Figure 2 shows examples of paired image reconstructions from the randomly sampled latent space of the fully supervised VAEVAE model. The digit next to the each reconstruction shows the digit class prediction for this image. The quantitative results in Figure 7 show that all three PoE based models reached a similar joint coherence as MMVAE, VAEVAE scored even higher. The cross-coherence

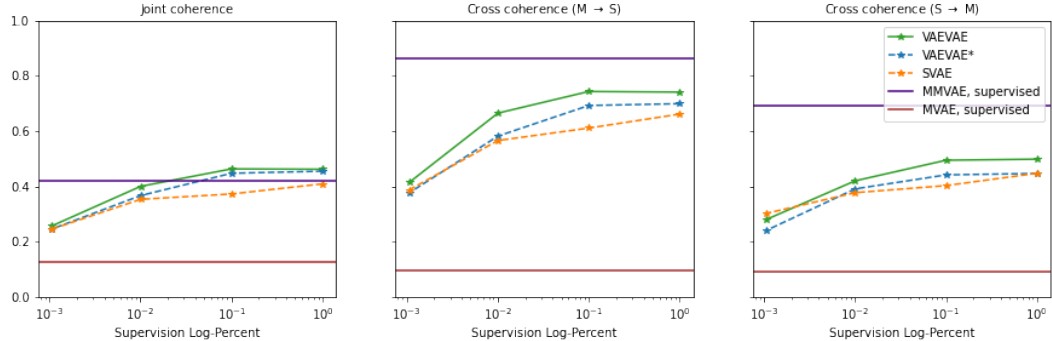

Figure 7: Performance on MNIST-SVHN for different supervision levels. (a) Joint coherence, a share of generated images with the same digit class; (b) Cross-coherence, accuracy of SVHN reconstructions given MNIST; (c) Cross-coherence, accuracy of MNIST reconstructions given SVHN.

results were best for MMVAE, but the three PoE based models performed considerably better than the MVAE baseline reported by Shi et al. (2019).

**Image and text: CUB-Captions** The second benchmark considered by Shi et al. (2019) is the CUB Images-Captions dataset (Wah et al., 2011) containing photos of birds and their textual descriptions. Here the modalities are of different nature but similar in dimensionality and information content. We used the source code[3] by Shi et al. to compute the same evaluation metrics as in the MMVAE study. Canonical correlation analysis (CCA) was used for estimating joint and cross-coherences of images and text (Massiceti et al., 2018). The projection matrices $W_x$ for images and $W_y$ for captions were pre-computed using the training set of CUB Images-Captions and are available as part of the source code. Given a new image-caption pair $\tilde{x}, \tilde{y}$, we computed the correlation between the two by $\mathrm{corr}(\tilde{x}, \tilde{y}) = \frac{\phi(\tilde{x})^T \phi(\tilde{y})}{\|\phi(\tilde{x})\|\|\phi(\tilde{y})\|}$, where $\phi(\tilde{k}) = W_k^T \tilde{k} - \mathrm{avg}(W_k^T k)$.

We employed the same image generation procedure as in the MMVAE study. Instead of creating the images directly, we generated 2048-d feature vectors using a pre-trained ResNet-101. In order to find the resulting image, a nearest neighbours lookup with Euclidean distance was performed. A CNN encoder and decoder was used for the (see Table F.5 and Table F.6). Prior to computing the correlations, the captions were converted to 300-d vectors using FastText (Bojanowski et al., 2017). As in the experiment before, we used the same network architectures and hyperparameters as Shi et al. (2019). We sampled 1000 latent space vectors from the prior distribution. Images and captions were then reconstructed with the decoding networks. The joint coherence was then computed as the CCA for the resulting image and caption averaged over the 1000 samples. Cross-coherence was computed from caption to image and vice versa using the CCA averaged over the whole test set.

As can be seen in Figure 8, VAEVAE showed the best performance among all models. With full supervision the VAEVAE model outperformed MMVAE in all three metrics. The cross-coherence of the three PoE models was higher or equal to MMVAE except for very low supervision levels. All three PoE based models were consistently better than MVAE.

## 5 DISCUSSION AND CONCLUSIONS

We studied bi-modal variational autoencoders (VAEs) based on a product-of-experts (PoE)architecture, in particular VAEVAE as proposed by Wu & Goodman (2019) and a new model SVAE, which we derived in an axiomatic way, and represents a generalization of the VAE-VAE architecture. The models learn representations that allow coherent sampling of the modalities and accurate sampling of one modality given the other. They work well in the semi-supervised setting, that is, not all modalities need to be always observed during training. It has been argued that the mixture-of-experts (MoE) approach MMVAE is preferable to a PoE for multimodal VAEs (Shi et al., 2019), in particular in the fully supervised setting (i.e., when all data are paired).

---

[3]https://github.com/iffsid/mmvae

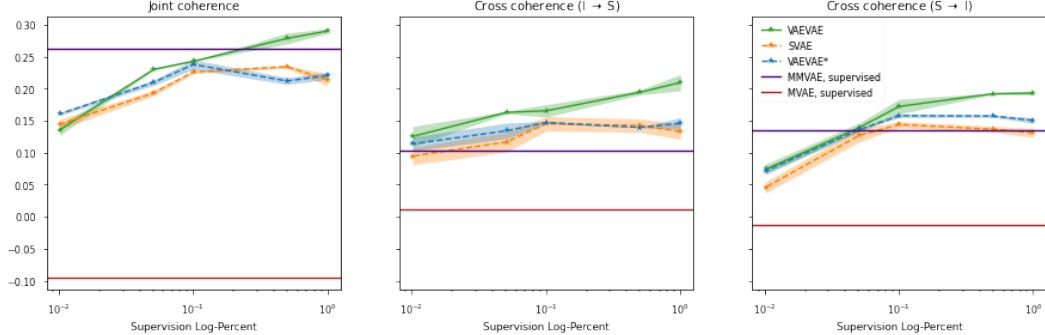

Figure 8: CUB Images-Captions dataset. Performance metrics for different supervision levels. (a) Joint coherence, the correlation between images and labels reconstructed from the randomly sampled latent vectors; (b) Cross-coherence, the correlation of the reconstructed caption given the image; (c) Cross-coherence, the correlation of the reconstructed image given the caption.

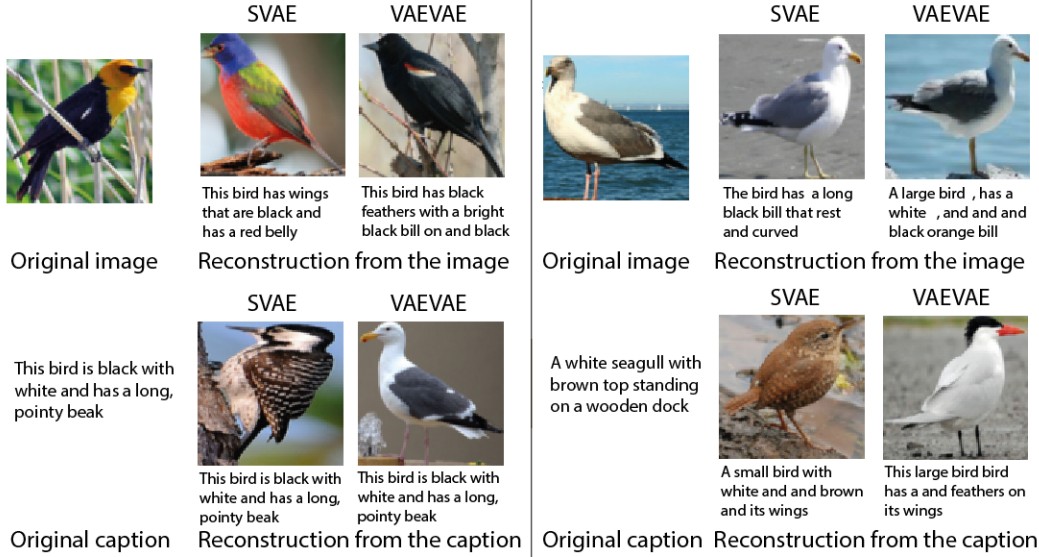

Figure 9: Examples of image and caption reconstructions given one modality input for SVAE and VAEVAE. Given that the caption can be broad (e.g., "this bird is black and white and has a long pointy beak" in the example), it can fit many different images. In this case, the image from the caption reconstruction tends to better fit the description than the original image. The same goes for images: one of the reconstructed images has a bird with a red belly which got reflected in the generated caption even though it was not a part of the original caption.

This conjecture was based on a comparison with the MVAE model (Wu & Goodman, 2018), but is refuted by our experiments showing that VAEVAE and our newly proposed SVAE can outperform MMVAE on experiments conducted by Shi et al. (2019). Intuitively, MoEs are more tailored towards an "OR" (additive) combination of the information provided by the different modalities, while PoEs are more tailored to towards an "AND" (multiplicative) combination. This is demonstrated by our experiments on halved digit images, where a conjunctive combination is helpful and the PoE models perform much better than MMVAE. We also expand SVAE and VAEVAE to 3-modal case and show that SVAE demonstrates better performance on individual modalities reconstructions while having less parameters than VAEVAE.

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

## A    DERIVATION OF THE MODEL ARCHITECTURE

We define our model in an axiomatic way, requiring the following properties:

1. When no modalities are observed, the generating distribution for the latent code is Gaussian:
$$q(z) = p(z) = \mathcal{N}(0, I) \tag{A.11}$$
This property is well known from VAEs and allows easy sampling.

2. The two modalities are independent given the latent code, so the decoder distribution is:
$$p(x_1, x_2|z) = p_1(x_1|z)p_2(x_2|z) \tag{A.12}$$

The second property formalizes our goal that the latent representation contains all relevant information from all modalities.

The joint distribution $p(z|x) = p(z|x_1, x_2)$ is given by

$$p(z|x_1, x_2) = \frac{p(z)p_1(x_1, x_2|z)}{p(x_1, x_2)} = \frac{p(z)p_1(x_1, x_2|z)}{\int p(z')p(x_1, x_2|z')\mathrm{d}z'}$$
$$\overset{(A.12)}{=} \frac{p(z)p_1(x_1|z)p_2(x_2|z)}{\int p(z')p(x_1|z')p(x_2|z')\mathrm{d}z'} \quad . \tag{A.13}$$

3. Both experts cover the whole latent space with equal probabilities:
$$q_1(z) = \int q_1(z|x_1)p(x_1)\mathrm{d}x_1 = \int q_2(z|x_2)p(x_2)\mathrm{d}x_2 = q_2(z) \tag{A.14}$$

4. The joint encoding distribution $q(z|x) = q(z|x_1, x_2)$ is assumed to be given by the product-of-experts rule (Hinton, 2002; Welling, 2007):
$$q(z|x_1, x_2) = \frac{1}{Z(x_1, x_2)}q_1(z|x_1)q_2(z|x_2) \tag{A.15}$$

with $Z(x_1, x_2) = \int q_1(z'|x_1)q_2(z'|x_2)\mathrm{d}z'$. The modelling by a product-of-experts in equation A.15 is a simplification of equation A.13 to make the model tractable.

Given equation A.15 and equation A.14 we obtain

$$q(z) = \int q(z|x)p(x)\mathrm{d}x = \int q(z|x_1, x_2)p(x_1, x_2)\mathrm{d}x_1\mathrm{d}x_2$$

$$\stackrel{(A.15)}{=} \int \frac{1}{Z(x_1, x_2)} q_1(z|x_1)q_2(z|x_2)p(x_1)p(x_2|x_1)\mathrm{d}x_1\mathrm{d}x_2 \quad . \quad (A.16)$$

Let us define

$$q_j^*(z|x_i) = \int \frac{1}{Z(x_i, x_j)} q_j(z|x_j)p(x_j|x_i)\mathrm{d}x_j \tag{A.17}$$

and write

$$q(z) \stackrel{(A.16)}{=} \int q_1(z|x_1)p(x_1) \int \frac{1}{Z(x_1, x_2)} q_2(z|x_2)p(x_2|x_1)\mathrm{d}x_2\mathrm{d}x_1$$

$$= \int p(x_1)q_1(z|x_1)q_2^*(z|x_1)\mathrm{d}x_1 \quad . \quad (A.18)$$

So the proposal distributions are:

$$q(z|x_1, x_2) = \frac{1}{Z(x_1, x_2)} q_1(z|x_1)q_2(z|x_2) \tag{A.19}$$

$$q(z|x_1) = q_1(z|x_1)q_2^*(z|x_1) \tag{A.20}$$

$$q(z|x_2) = q_2(z|x_2)q_1^*(z|x_2) \tag{A.21}$$

$$q(z) = \mathcal{N}(0, I) \tag{A.22}$$

## B DERIVATION OF THE LOSS FUNCTION

In the following, we derive the ELBO-type loss we use for training, see equation 7. Let consider the optimization

$$E_{p_{\mathrm{Data}}(x_1,x_2)}[\log p(x_1, x_2)] = \frac{1}{2}E_{p_{\mathrm{Data}}(x_1,x_2)}[\log p(x_1|x_2) + \log p(x_2) + \log p(x_2|x_1) + \log p(x_1)]$$

$$= \frac{1}{2}E_{p_{\mathrm{Data}}(x_1,x_2)}[\log p(x_1|x_2)] + \frac{1}{2}E_{p_{\mathrm{Data}}(x_1,x_2)}[\log p(x_2|x_1)]$$

$$+ \frac{1}{2}E_{p_{\mathrm{Data}}(x_2)}[\log p(x_2)] + \frac{1}{2}E_{p_{\mathrm{Data}}(x_1)}[\log p(x_1)] \quad . \quad (B.23)$$

We can now proceed by finding lower-bounds for each term. For the last two terms $\log p(x_i)$ we can use the standard ELBO as given in equation 1. This gives the terms

$$\mathcal{L}_i = E_{p_{\mathrm{Data}}(x_i)}\left[E_{q(z|x_i)}[\log p_i(x_i|z)] - D_{\mathrm{KL}}(q(z|x_i) \| p(z))\right] \tag{B.24}$$

Next, we will derive $\log p(x_1|x_2)$. This we can do in terms of a conditional VAE (Sohn et al., 2015), where we condition all terms on $x_2$ (or $x_1$ if we model $\log p(x_2|x_1)$). So the model we derive the log-likelihood for is $p(x_1|x_2) = \int p(x_1|z)p(z|x_2)dz$, where $p(z|x_2)$ is now our prior. By model assumption we further have $p(x_1, x_2, z) = p(x_1|z)p(x_2|z)p(z)$ and therefore $p(x_1|x_2, z) = p(x_1|z)$. Thus we arrive at the ELBO losses

$$\mathcal{L}_{12} = E_{p_{\mathrm{Data}}(x_1,x_2)}\left[E_{q(z|x_1,x_2)}[\log p_1(x_1|z)] - D_{\mathrm{KL}}(q(z|x_1, x_2) \| p(z|x_2))\right] \tag{B.25}$$

and

$$\mathcal{L}_{21} = E_{p_{\mathrm{Data}}(x_1,x_2)}\left[E_{q(z|x_1,x_2)}[\log p_2(x_2|z)] - D_{\mathrm{KL}}(q(z|x_1, x_2) \| p(z|x_1))\right] \quad . \quad (B.26)$$

We now insert the terms in equation B.23 and arrive at:

$$2E_{p_{\mathrm{Data}}(x_1,x_2)}[\log p(x_1, x_2)] \leq \mathcal{L}_{12} + \mathcal{L}_{21} + \mathcal{L}_1 + \mathcal{L}_2$$

$$= E_{p_{\mathrm{Data}}(x_1,x_2)}\left[E_{q(z|x_1,x_2)}[\log p_1(x_1|z)] - D_{\mathrm{KL}}(q(z|x_1, x_2) \| p(z|x_2))\right]$$

$$+ E_{p_{\mathrm{Data}}(x_1,x_2)}\left[E_{q(z|x_1,x_2)}[\log p_2(x_2|z)] - D_{\mathrm{KL}}(q(z|x_1, x_2) \| p(z|x_1))\right]$$

$$+ E_{p_{\mathrm{Data}}(x_1)}\left[E_{q(z|x_1)}[\log p_1(x_1|z)] - D_{\mathrm{KL}}(q(z|x_1) \| p(z))\right]$$

$$+ E_{p_{\mathrm{Data}}(x_2)}\left[E_{q(z|x_2)}[\log p_2(x_2|z)] - D_{\mathrm{KL}}(q(z|x_2) \| p(z))\right]$$

The first two terms together give

$$E_{p_{\text{Data}}(x_1,x_2)}\left[E_{q(z|x_1,x_2)}[\log p_1(x_1|z) + \log p_2(x_2|z)]\right] \quad . \tag{B.27}$$

We do not know the conditional prior $p(z|x_i)$. By definition of the VAE, we are allowed to optimize the prior, therefore we can parameterize it and optimize it. However, we know that in an optimal model $p(z|x_i) \approx q(z|x_i)$ and it might be possible to prove that if $p(z|x_i)$ is learnt in the same model-class as $q(z|x_i)$ we can find that the optimum is indeed $p(z|x_i) = q(z|x_i)$. Inserting this choice into the equation gives the end-result.

## C  TRAINING PROCEDURE FOR SVAE AND VAEVAE*

---

**Algorithm 1:** Training procedure for SVAE and VAEVAE*. In bold are terms that are different from Wu & Goodman (2019)

---

**Input:** Supervised example $(x_1, x_2)$, Unsupervised example $x_1'$, Unsupervised example $x_2'$
$z' = q(z|x_1, x_2)$
$z_{x_1} = q_1(z|x_1)$
$z_{x_2} = q_2(z|x_2)$
$d_1 = D_{\text{KL}}(q(z'|x_1, x_2)\|q_1(z_{x_1}|x_1)) + \boldsymbol{D_{\text{KL}}(q_1(z_{x_1}|x_1)\|p(z))}$
$d_2 = D_{\text{KL}}(q(z'|x_1, x_2)\|q_2(z_{x_2}|x_2)) + \boldsymbol{D_{\text{KL}}(q_2(z_{x_2}|x_2)\|p(z))}$
$\mathcal{L} = \log p_1(x_1|z) + \log p_2(x_2|z) + \boldsymbol{\log p_1(x_1|z_{x_1})} + \boldsymbol{\log p_2(x_2|z_{x_2})} + d_1 + d_2$
$\mathcal{L}_{x_1} = \log p_1(x_1'|z_{x_1}) + D_{\text{KL}}(q_1(z_{x_1}|x_1')\|p(z))$
$\mathcal{L}_{x_2} = \log p_2(x_2'|z_{x_2}) + D_{\text{KL}}(q_2(z_{x_2}|x_2')\|p(z))$
$\mathcal{L}_{\text{comb}} = \mathcal{L}' + \mathcal{L}_{x_1} + \mathcal{L}_{x_2}$

---

## D  SVAE AND VAEVAE FOR MORE THAN TWO MODALITIES

In the following, we formalize the VAEVAE model for three modalities and present a naïve extension of the SVAE model to more than two modalities.

In the canonical extension of VAEVAE to three modalities, the three- and two-modal relations are captured by the corresponding networks $q(z|x_1, x_2, x_3)$, $q(z|x_i, x_j)$ and $q(z|x_i)$ for $i, j \in \{1, 2, 3\}$, see Figure 5. In the general $n$-modal case, the model has $2^n$ networks. For $n = 3$, the loss function reads:

$$
\begin{aligned}
\mathcal{L}_{1,2,3} =& E_{p_{\text{paired}}(x_1,x_2,x_3)}\left[E_{q(z|x_1,x_2,x_3)}[\log p_1(x_1|z) + \log p_2(x_2|z) + \log p_3(x_3|z)]\right]\\
& - D_{\text{KL}}(q(z|x_1, x_2, x_3) \| q(z|x_1, x_2))\\
& - D_{\text{KL}}(q(z|x_1, x_2, x_3) \| q(z|x_2, x_3))\\
& - D_{\text{KL}}(q(z|x_1, x_2, x_3) \| q(z|x_1, x_3)) \tag{D.28}\\
\mathcal{L}_{ij} =& E_{p_{\text{paired}}(x_1,x_2,x_3)}\left[E_{q(z|x_i,x_j)}[\log p_i(x_i|z) + \log p_j(x_j|z)]\right]\\
& - D_{\text{KL}}(q(z|x_i, x_j) \| q(z|x_1)) - D_{\text{KL}}(q(z|x_i, x_j) \| q(z|x_2))\\
& - D_{\text{KL}}(q(z|x_i, x_j) \| q(z|x_3)) - D_{\text{KL}}(q(z|x_i, x_j) \| q(z)) \tag{D.29}\\
\mathcal{L}_i =& E_{p_{\text{unpaired}}(x_i)}\left[E_{q(z|x_i)}[\log p_i(x_i|z)] - D_{\text{KL}}(q(z|x_i) \| q(z))\right] \tag{D.30}\\
\mathcal{L}_{\text{comb}} =& \mathcal{L}_{1,2,3} + \sum_{i,j\in\{1,2,3\},i\neq j} \mathcal{L}_{i,j} + \sum_{i=1}^{3} \mathcal{L}_i \tag{D.31}
\end{aligned}
$$

In this study, we considered a simplifying extension of SVAE to $n$ modalities using $n^2$ networks $q_i^j(z|x_j)$ for $i, j \in \{1, \dots, n\}$. For the 3-modal case depicted in Figure 5, the PoE relations between

the modalities are defined in the following way:

$$q(z|x_1, x_2, x_3) = \frac{1}{Z(x_1, x_2, x_3)} q_1^1(z|x_1) q_2^2(z|x_2) q_3^3(z|x_3) \tag{D.32}$$

$$i, j, k \in \{1, 2, 3\}, i \neq j \neq k : \tag{D.33}$$

$$q^i(z|x_i, x_j) = \frac{i}{Z(x_i, x_j)} q_i^i(z|x_i) q_j^j(z|x_j) q_k^i(z|x_i) \tag{D.34}$$

$$q^j(z|x_i, x_j) = \frac{1}{Z(x_i, x_j)} q_i^i(z|x_i) q_j^j(z|x_j) q_k^j(z|x_j) \tag{D.35}$$

$$q(z|x_i) = q_i^i(z|x_i) q_j^i(z|x_i) q_k^i(z|x_i) \tag{D.36}$$

$$q(z) = \mathcal{N}(0, I) \tag{D.37}$$

The corresponding SVAE loss function has additional terms due to the fact that the relations between pairs of modalities need to be captured with two PoE rules $q^i(z|x_i, x_j)$ and $q^i(z|x_i, x_j)$ in SVAE, while there is only a single network $q(z|x_i, x_j)$ in VAEVAE. The loss functions equation D.28–equation D.31 above are modified in a way that $f(q(z|x_i, x_j)) = f(q^i(z|x_i, x_j)) + f(q^j(z|x_i, x_j))$ for any function $f$.

This extension of the bi-modal case assumes that $p(x_i, x_j|x_k) = p(x_i|x_k)p(x_j|x_k)$ for $i, j, k \in \{1, 2, 3\}, i \neq j \neq k$, which implies that $x_i$, $x_j$ and $x_k$ are independent of each other.

## E    QUALITATIVE EXAMPLES

### E.1    SVAE

See Figure E.10

### E.2    VAEVAE

See Figure E.11

## F    TRAINING DETAILS AND ENCODER/DECODER ARCHITECTURES

The encoder and decoder architectures for each experiment and modality are listed below. To implement joint encoding network (VAEVAE architecture), an fully connected layer followed by ReLU is added to the encoding architecture for each modality. Another fully connected layer accepts the concatenated features from the two modalities as an input and outputs the latent space parameters. Adam optimiser is used for learning in all the models (Kingma & Ba, 2015).

### F.1    MNIST-SPLIT

The models are trained for 200 epochs with the learning rate $2 \cdot 10^{-4}$. The best epoch is chosen by the highest accuracy of the reconstruction from the top half evaluated on the validation set. We used a latent space dimensionality of $L = 64$. The network architectures are described in Table F.7.

| Encoder | Decoder |
|---|---|
| Input $\in \mathbb{R}^{3x32x32}$ | Input $\in \mathbb{R}^L$ |
| 4x4 conv. 64 stride 2 pad 1 & ReLU | FC. L & ReLU |
| 4x4 conv. 128 stride 2 pad 1 & ReLU | FC. 512 & ReLU |
| 4x4 conv. 256 stride 2 pad 1 & ReLU | FC. 112 & ReLU |
| FC. 786 & ReLU | 4x4 upconv. 56 stride 1 pad 0 & ReLU |
| FC. L, FC. L | 4x4 upconv. 28 stride 2 pad 1 & ReLU |

Table F.2: Network architectures for MNIST-Split for each image half.

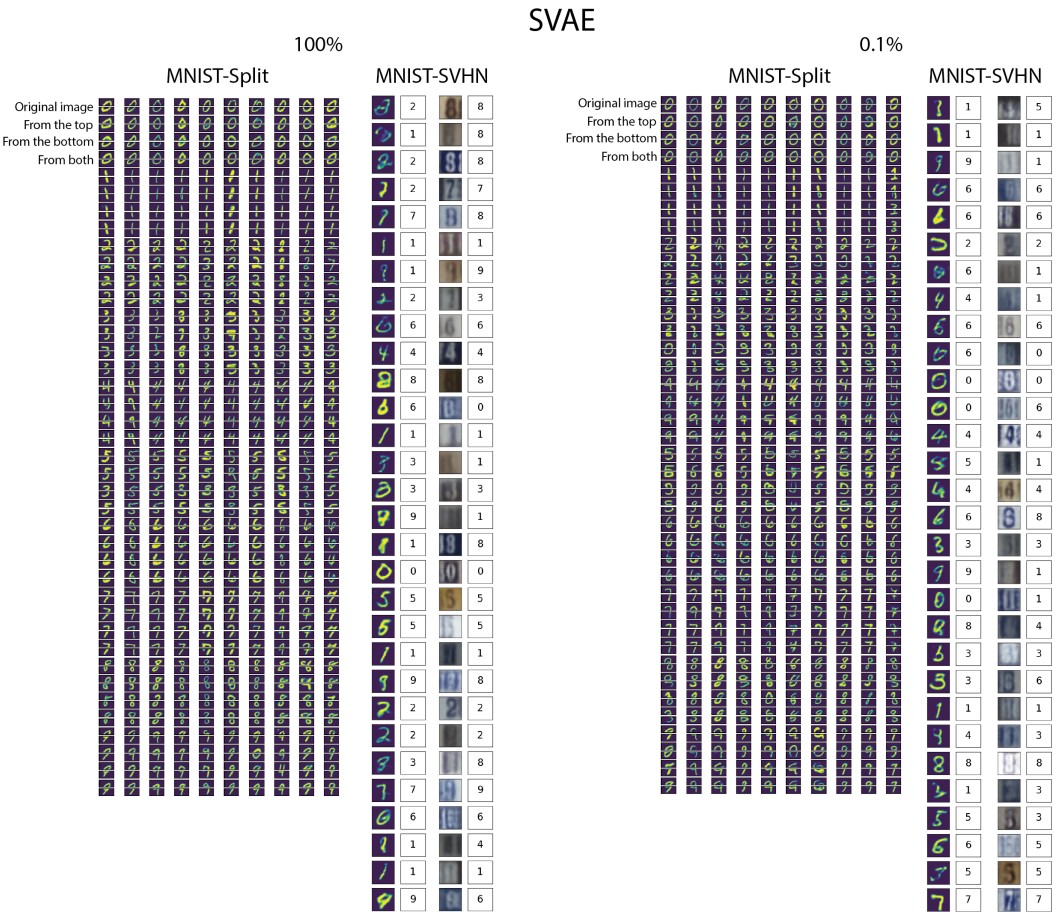

Figure E.10: A. MNIST-Split image reconstructions of a top half and a bottom half given (a) the top half; (b) the bottom half of the original image (c) both halves. B. Side-by-side MNIST-SVHN reconstruction from randomly sampled latent space, with oracle predictions of a digit class. The joint coherence from the Figure 7 is a share of classes predicted the same. The examples are generated by SVAE for the supervision levels 100% and 0.1%

## F.2  MNIST-SVHN

The models are trained for 50 epochs with the learning rate $10^{-3}$ for VAEVAE and VAEVAE* and $10^{-4}$ for SVAE. The best epoch is chosen by the highest joint coherence evaluated on the validation set. We used a latent space dimensionality of $L = 20$. The network architectures are summarized in Table F.4 and Table F.3 for the MNIST and SVHN modality, respectively.

| Encoder | Decoder |
|---|---|
| Input $\in \mathbb{R}^{1x28x28}$ | Input $\in \mathbb{R}^L$ |
| FC. 400 & ReLU | FC. 400 & ReLU |
| FC. $L$, FC. $L$ | FC. 1 x 28 x 28 Sigmoid |

Table F.3: Network architectures for MNIST-SVHN: MNIST.

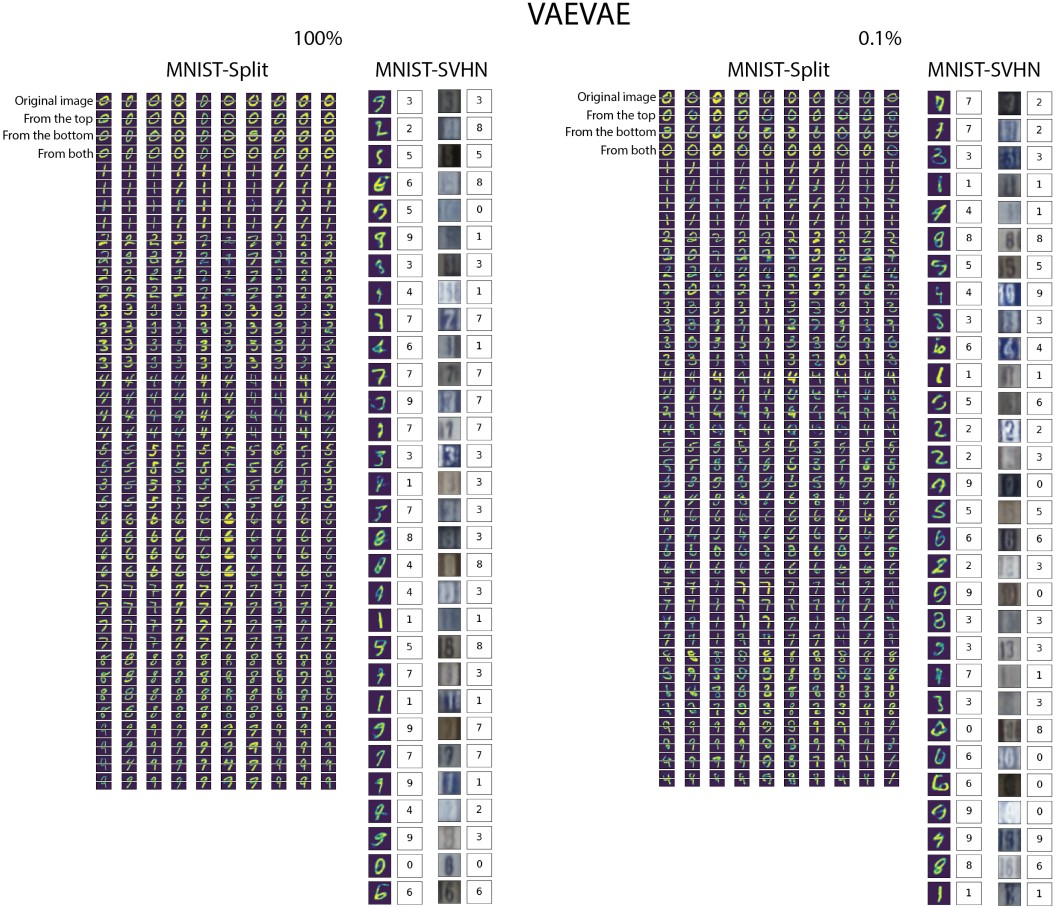

Figure E.11: A. MNIST-Split image reconstructions of a top half and a bottom half given (a) the top half; (b) the bottom half of the original image (c) both halves. B. Side-by-side MNIST-SVHN reconstruction from randomly sampled latent space, with oracle predictions of a digit class. The joint coherence from the Figure 7 is a share of classes predicted the same. The examples are generated by VAEVAE for the supervision levels 100% and 0.1%

## F.3 CUB-CAPTIONS

The models are trained for 200 epochs with the learning rate $10^{-4}$. The best epoch is chosen by the highest joint coherence evaluated on the validation set. We used a latent space dimensionality of $L = 64$. The network architectures are described in Table F.5 and Table F.6 for the text and image modality, respectively.

## F.4 MNIST-SPLIT-THREE

The models are trained for 50 epochs with the learning rate $2 \cdot 10^{-4}$. The best epoch is chosen by the highest accuracy of the reconstruction from the top half evaluated on the validation set. We used a latent space dimensionality of $L = 64$. The network architectures are described in Table F.7.

| **Encoder** |
| --- |
| Input $\in \mathbb{R}^{3x32x32}$ |
| 4x4 conv. 32 stride 2 pad 1 & ReLU |
| 4x4 conv. 64 stride 2 pad 1 & ReLU |
| 4x4 conv. 128 stride 2 pad 1 & ReLU |
| 4x4 conv. $L$ stride 1 pad 0, 4x4 conv. $L$ stride 1 pad 0 |

| **Decoder** |
| --- |
| Input $\in \mathbb{R}^{L}$ |
| 4x4 upconv. 128 stride 1 pad 0 & ReLU |
| 4x4 upconv. 64 stride 2 pad 1 & ReLU |
| 4x4 upconv. 32 stride 2 pad 1 & ReLU |
| 4x4 upconv. 3 stride 2 pad 1 & Sigmoid |

Table F.4: Network architectures for MNIST-SVHN: SVHN.

| **Encoder** |
| --- |
| Input $\in \mathbb{R}^{1590}$ |
| Word Emb. 256 |
| 4x4 conv. 32 stride 2 pad 1 & BatchNorm2d & ReLU |
| 4x4 conv. 64 stride 2 pad 1 & BatchNorm2d & ReLU |
| 4x4 conv. 128 stride 2 pad 1 & BatchNorm2d & ReLU |
| 1x4 conv. 256 stride 1x2 pad 0x1 & BatchNorm2d & ReLU |
| 1x4 conv. 512 stride 1x2 pad 0x1 & BatchNorm2d & ReLU |
| 4x4 conv. $L$ stride 1 pad 0, 4x4 conv. $L$ stride 1 pad 0 |

| **Decoder** |
| --- |
| Input $\in \mathbb{R}^{L}$ |
| 4x4 upconv. 512 stride 1 pad 0 & ReLU |
| 1x4 upconv. 256 stride 1x2 pad 0x1 & BatchNorm2d & ReLU |
| 1x4 upconv. 128 stride 1x2 pad 0x1 & BatchNorm2d & ReLU |
| 4x4 upconv. 64 stride 2 pad 1 & BatchNorm2d & ReLU |
| 4x4 upconv. 32 stride 2 pad 1 & BatchNorm2d & ReLU |
| 4x4 upconv. 1 stride 2 pad 1 & ReLU |
| Word Emb.$^{\mathrm{T}}$ 1590 |

Table F.5: Network architectures for CUB-Captions language processing.

| **Encoder** | **Decoder** |
| --- | --- |
| Input $\in \mathbb{R}^{2048}$ | Input $\in \mathbb{R}^{L}$ |
| FC. 1024 ELU | FC. 256 ELU |
| FC. 512 ELU | FC. 512 ELU |
| FC. 256 ELU | FC. 1024 ELU |
| FC. L, FC. L | FC. 2048 |

Table F.6: Network architectures for CUB-Captions image processing.

| **Encoder** | **Decoder** |
| --- | --- |
| Input $\in \mathbb{R}^{3x32x32}$ | Input $\in \mathbb{R}^{L}$ |
| 4x4 conv. 64 stride 2 pad 1 & ReLU | FC. L & ReLU |
| 4x4 conv. 128 stride 2 pad 1 & ReLU | FC. 512 & ReLU |
| 4x4 conv. 256 stride 2 pad 1 & ReLU | FC. 112 & ReLU |
| FC. 786 & ReLU | 4x4 upconv. 56 stride 1 pad 0 & ReLU |
| FC. L, FC. L | 2x4 upconv. 28 stride 2 pad 1 & ReLU |

Table F.7: Network architectures for MNIST-Split-Three for each image part.

