# OpenReview forum: "Multimodal Variational Autoencoders for Semi-Supervised Learning: In Defense of Product-of-Experts"
_ICLR.cc/2021/Conference — Reject_

### Official Review · AnonReviewer1 · 2020-10-27

**Rating:** 5
**Confidence:** 3

**Review:**

Pros:
This paper proposed SVAE as a more general form for the previous VAEVAE model.

Cons:
1). In the subsection "Image and text: CUB-caption", the definition of \phi(\tilt{k}) is a little bit confusing. Is there a typo (\tilt{k} == k), or what is k, a constant?
2). From all the experiments listed in the paper, we can see that VAEVAE performs best. And the content is lack of analysis why VAEVAE is better than SVAE in practice. From the subsection "SVAE vs VAEVAE" we know that SVAE is a general form of VAEVAE, but it becomes less important since the general form performs worse than a special case in practice. In other words, the importance of the works is not that convinicing. At least, as a reader, we expect to see in some cases, SVAE is better than VAEVAE.
3) In equation (10), will weighted sum be a better choice?

---

> ### Author Response · Authors · 2020-11-24
> **Multimodal learning; SVAE advantages; loss function definition**
>
> We thank the reviewer for the helpful comments.
>
> >In the subsection "Image and text: CUB-caption", the definition of \phi(\tilt{k}) is a little bit confusing. Is there a typo (\tilt{k} == k), or what is k, a constant?
>
> The \tilt{k} corresponds to one sample, while k is a training set of samples. After revising the definition we however did fix another typo - corr(x, y) should be corr(\tilt{x}, \tilt{y})
>
> >From all the experiments listed in the paper, we can see that VAEVAE performs best. And the content is lack of analysis why VAEVAE is better than SVAE in practice. From the subsection "SVAE vs VAEVAE" we know that SVAE is a general form of VAEVAE, but it becomes less important since the general form performs worse than a special case in practice. In other words, the importance of the works is not that convinicing. At least, as a reader, we expect to see in some cases, SVAE is better than VAEVAE.
>
> We added another experiment where SVAE performs better than VAEVAE for reconstructing individual modalities, see Figure 6.
> We would like to point at several aspects of SVAE model that are beneficial: 1) In the latest revision of the paper we explored the 3 modalities scenario using the MNIST-Split-3 experiment and showed that SVAE better reconstructs the full image from individual modalities. 2) While the number of parameters is the same for bi-modal SVAE and VAEVAE, with n modalities the number of parameters grows exponentially for VAEVAE, while being in order of n^2 for our simple extension of SVAE. 3) The conceptual difference between SVAE and VAEVAE architectures and some of the experiment results (MNIST-Split <1% supervision) suggest that SVAE can better utilise the supervised data in a very low supervision level case. 4) SVAE is based on PoEs, thus is the most illustrative in the context of the paper.
>
> >In equation (10), will weighted sum be a better choice?
>
> Thank you for suggesting this improvement. We agree that optimizing the weights for the supervised/unsupervised loss might improve model performance. Since the focus of the paper is on comparing different models, we kept the individual model hyperparameter optimization at minimum, but it could be an important step for reaching state-of-art reconstruction for a real application.

---

### Official Review · AnonReviewer4 · 2020-10-28
**A discussion on PoE-based VAEs versus MoE-based ones that can be strenghtened by deeper experiments and discussions**

**Rating:** 4
**Confidence:** 4

**Review:**

# Update
I thank the authors for their comments and answers. While they agree on some of the concerns I raised, others are still left open.
I believe they could be addressed in new major revision of the paper and I encourage authors to do so.

# Summary
This paper qualifies as a discussion around architectural choices when using variational-autoencoders (VAEs) for multi-modal learning.
The main claim is that architectures supporting the mixture-of-experts (MoEs) paradigm favor benchmarks where modalities appear in an 'OR'-relationship, while those implementing products-of-experts (PoEs) favor 'AND'-relationships.
To show this point and in 'defense' of PoE-VAEs, the paper introduces a missing-value imputation experiment over MNIST images to evaluate the second aspect.
As another contribution, the authors introduce the SVAE as a variation of a sota PoE-VAE, the VAEVAE, for bi-modal learning. Specifically, they introduce a variation that employs auxiliary encoder networks for each modality and derive accordingly a new composite ELBO loss to train them.
%
The point raised by author is reasonable, but the execution of the experimental section and little relevance of the introduced SVAE limit the value of the paper in its current state. Detailed comments follow.


# Presentation
The paper is generally understandable and well-written.
The introduction might benefit a rewriting as to gently introduce the larger scope of multimodal learning and then focusing on the more recent advancements, such as PoE-VAEs and MoE-VAEs.
For example, the first lines of the first paragraph directly jump to PoEs without providing the reader enough background.

I suggest authors to clarify that the scenario they are tackling is that of generative modeling in presence of missing data on the _inputs_ (as a modality missing is a special case of missing not at random) more than referring to it as semi-supervised learning.
I recognize that the authors follow a recent trend in the VAE-literature, but classically, semi-supervised has been referred to as the case when missing values are on the _output_, in a clear discriminative setting.

Some typos are present, e.g., "which corresponds to ”AND” combination of the modalities and favors the MoE architecture" (should be "OR"). Clearly, they can be easily fixed by an additional proofreading pass.

# Contributions
On the one hand, the discussion of the pros and cons of PoE-VAEs and MoE-VAEs boils down to one conjecture ('AND' versus 'OR' modality fusion) that is shallowly tested in the experimental section.
Which can be greatly strengthened, see comments below.

On the other hand, the introduction of the SVAE architecture, while potentially appealing, seems a minor contribution after seeing that VAEVAE and MMVAE generally outperform it in the graphs and tables in the experimental section (with the exception of the first experiment, joint modalities and one single percentage of low supervision).
I would advice authors to explicitly say what is the benefit (maybe didactic?) of introducing the SVAE as a new model.

Furthermore, a limitation of a certain regard concerns dealing with bi-modal data only in the text and in the experiments.
This makes unclear what is the price to pay to scale these architectures to truly-multimodal data.
For examples, Eqs. 7-1 suggest that the new composite ELBO can be extended to include uni-modal ELBO terms for each modality. However, one ordering over modalities for conditioning (according to Appendix B) shall be chosen, and it is not clear how this can influence learning and inference (in a similar fashion variable ordering influences autoregressive models).
Architecture-wise, it is not self-evident how many additional encoder components are needed for more than two modalities. If more than one per modality, then the challenges should be discussed in depth.

# Experiments
As already stated, experiments are limited to bi-modal data only.
One additional downside of the experiments is that only coherence inter-modalities is measured as a metric. Sample/modality quality is not discussed, not even reported in a qualitative way (for the exception for some samples for the CUB dataset and some MNIST image reconstructions only for SVAE in Fig.3).
I suggest authors to report the FID scores (or any other suitable variant like KID or precision-recall curves) of the joint and single modalities for the generated images to assess their quality. This is a fundamental aspect as modalities can be coherent but very far from the true data distribution or still not exactly close to the reconstruction, which is just a mode of the whole distribution.
Along this direction, one shall evaluate conditional sampling and not only reconstructions to see if the VAE have collapsed to pointwise densities.

Lastly, the CUB experiment provides some empirical evidence that is hard to evaluate or pose in the context of generative modelling.
In order to follow the MMVAE paper, the authors are decoding images not in pixel space, but in the latent space of a  ResNet-101. Then the showed images are the nearest neighbours in the training data.
This is the opposite of the generative modelling paradigm, and misleading: equivalently accurate and good-looking final images could come from a model memorizing the training set.
I advice authors to evaluate the quality of generated samples and reconstructions in the pixel space (with the metrics discussed above), alternatively, to introduce a proper decoder for the ResNet-101 embedding or not to include the above experiment at all.

---

> ### Author Response · Authors · 2020-11-24
> **Multimodal learning; SVAE advantages; qualitative evaluation**
>
> We thank the reviewer for the helpful comments.
>
> >The introduction might benefit a rewriting as to gently introduce the larger scope of multimodal learning and then focusing on the more recent advancements, such as PoE-VAEs and MoE-VAEs. For example, the first lines of the first paragraph directly jump to PoEs without providing the reader enough background.
>
> Thank you for the feedback on the clarity of the manuscript. Following your advice, we restructured the introduction slightly, walking the reader through the multimodal generative modeling concept before introducing the PoE based models.
>
> >I suggest authors to clarify that the scenario they are tackling is that of generative modeling in presence of missing data on the inputs (as a modality missing is a special case of missing not at random) more than referring to it as semi-supervised learning.
>
> We agree. In referring to missing data as semi-supervised case we follow the previous work e.g. MVAE and VAEVAE [Wu et al.]. In these papers the “missing data” term is used alongside the “semi- or weakly- supervised” term. Referring to a share of missing inputs as a “supervision level” allows us to present the accuracies in the same way as in VAEVAE [Wu et al.], Figure 1. We added a comment on the standard use of “semi-supervised learning” on the first page.
>
> >I would advice authors to explicitly say what is the benefit (maybe didactic?) of introducing the SVAE as a new model.
> Architecture-wise, it is not self-evident how many additional encoder components are needed for more than two modalities. If more than one per modality, then the challenges should be discussed in depth.
>
> Thank you for encouraging us to explore the multi-modal capabilities of SVAE and VAEVAE models. We extended the manuscript with a section describing the extensions of the architectures to more than two modalities and with the new MNIST-Split-3 experiment, please see the new section “Image and image: MNIST-Split” starting on page 5, Figure 5 and 6,  and Appendix D.  Exploring the 3-modal case revealed that 1) a simple extension of SVAE scales better than VAEVAE, a number of parameters of which is growing exponentially for n modalities, while staying in order of n^2 for SVAE. 2) SVAE shows better performance for individual modalities in the 3-modalities experiment we conducted.
>
> The other benefits of SVAE are: 3) the conceptual difference between SVAE and VAEVAE architectures and some of the experiment results (MNIST-Split <1% supervision) suggest that SVAE can better utilise the supervised data in a low supervision level case. 4) SVAE heavily relies on PoEs and thus is the most illustrative in the context of the paper.
>
> >One additional downside of the experiments is that only coherence inter-modalities is measured as a metric. Sample/modality quality is not discussed, not even reported in a qualitative way (for the exception for some samples for the CUB dataset and some MNIST image reconstructions only for SVAE in Fig.3). I suggest authors to report the FID scores (or any other suitable variant like KID or precision-recall curves) of the joint and single modalities for the generated images to assess their quality.
>
> In order to demonstrate that the reconstructions match the input distributions, we added the supplementary section E with 100+ qualitative examples of the VAEVAE and SVAE reconstructions for very high and very low supervision levels.
>
> >Lastly, the CUB experiment provides some empirical evidence that is hard to evaluate or pose in the context of generative modelling. In order to follow the MMVAE paper, the authors are decoding images not in pixel space, but in the latent space of a ResNet-101. Then the showed images are the nearest neighbours in the training data. This is the opposite of the generative modelling paradigm, and misleading: equivalently accurate and good-looking final images could come from a model memorizing the training set.
>
> With multi-modal VAEs comparison being the main focus of the paper (our research on multi-modal VAEs is actually driven by an application not involving images), we prioritised comparability with the previous work when choosing the experiments, so we followed the experiments from MMVAE study. While we agree that the nearest neighbour search is not a full reconstruction, the investigated coherence metrics are still representative for the quality of the bi-modal joint and marginal learning.

---

### Official Review · AnonReviewer3 · 2020-10-28
**Comparing product-of-experts against mixture-of-experts in VAEs. Results don't seem conclusive enough.**

**Rating:** 4
**Confidence:** 4

**Review:**

This paper discusses and evaluates different generative models for multimodal data. Specifically, the authors are interested in comparing product-of-experts (PoE) to mixture-of-experts (MoE) in VAEs as ways to handle multimodality.

They also propose a novel model (SVAE) built on the PoE approach that, compared to previous models, aims to better handle missing modalities. In particular, they introduce additional networks that estimate the marginal distributions of the latent representations of the missing modalities given the observed ones.

They then evaluate multiple models on three tasks that highlight the difference of behavior in PoE and MoE-based models, concluding that PoEs are useful to model "AND" relationships in a multimodal setting.


################################################

Strong points:

-The paper is easy to follow, clearly presents the context and the different approaches. The proposed model is also elegantly designed and presented, with differences with previous methods highlighted.

-The authors seem to have provided most of the information needed for reproducibility.


Weaknesses:

-While SVAE is useful in the context of the paper as it uses explicit PoEs, the experiments don't show conclusive differences in performances with VAEVAE and VAEVAE* variants, reducing the potential impact of the novel elements of the paper. Actually, VAEVAE* (that is very close to the prior existing VAEVAE) obtains better or comparable performances except on the MNIST-split task with 1% or less paired data.
Also, uncertainty bars in the plots would be very useful, especially since there seem to be very large variations in what should theoretically be mostly monotonous curves (Figure 4 and Figure 6 left).

-The MNIST-split results are used to support the idea that PoEs are better-suited to model "AND" relationships (both modalities carry complementary information and are needed for the task). However, it seems that a system (composed of VAEVAE and the oracle) that takes as input only the top part is able to predict the correct class with an accuracy of 0.887. That tends to indicate that the evaluated task is fundamentally an "OR" task, and therefore ill-suited to evaluate the adequacy of the method to capture "AND" relationships. Can the authors comment on this potential issue?

-The paper is articulated around proving that PoE models can obtain good performances. Since VAEVAE already performs well and that the opposite view is only supported by a single publication, the importance of this particular contribution is questionable.


################################################

Score motivation:

Mainly, I believe the shown results aren't conclusive enough. They can't support the proposed model, nor the other insights of the paper.


################################################

Other question:

Can the authors comment on why MMVAE is terrible at auto-encoding an MNIST image (0.539 accuracy in table 1)? It seems very low.



################################################

################################################

Post-Rebuttal Update:

I'd like to thank the authors for their updates, the additional experiments are especially welcome. After taking into consideration the responses and the new evidence, I believe the concerns I raised still stand. Therefore, I keep the previous rating.

---

> ### Author Response · Authors · 2020-11-24
> **SVAE advantages; PoE vs MoE tasks**
>
> We thank the reviewer for the helpful comments
>
> >While SVAE is useful in the context of the paper as it uses explicit PoEs, the experiments don't show conclusive differences in performances with VAEVAE and VAEVAE* variants, reducing the potential impact of the novel elements of the paper.
>
> The newly added experiments on three modalities show a conceptual advantage of SVAE compared to VAEVAE as well as better performance of the newly proposed method, please see the new figures 5 and 6.
> For two modalities, our experimental results provide no arguments for preferring  SVAE over VAEVAE.
>
> >Uncertainty bars in the plots would be very useful, especially since there seem to be very large variations in what should theoretically be mostly monotonous curves (Figure 4 and Figure 6 left).
>
> We have added the error bars to the mentioned figures.
>
> >The MNIST-split results are used to support the idea that PoEs are better-suited to model "AND" relationships (both modalities carry complementary information and are needed for the task). However, it seems that a system (composed of VAEVAE and the oracle) that takes as input only the top part is able to predict the correct class with an accuracy of 0.887. That tends to indicate that the evaluated task is fundamentally an "OR" task, and therefore ill-suited to evaluate the adequacy of the method to capture "AND" relationships.
>
> Yes, MNIST-Split is not 100% an “AND” task. In many cases one can guess the right digit by just seeing the upper or lower half of it. Thus, given enough training data, a reasonably good performance can be expected in this scenario. For our study, it is important that it is more of an “AND” task than MNIST-SVHN, which is clearly an “OR” task. We could have designed an artificial pure “AND” task, but did not do so because (1) we do not think that this is very realistic in practice and (2) cross-coherence experiments would be meaningless in this case. Note that our concept of “AND” and “OR” tasks implicitly assumes a third modality (a label, the digit class in our experiments).
> We revised our text and hope that it is more clear now, see page 5 (2nd and 4th paragraph of Section 4).
>
> >Can the authors comment on why MMVAE is terrible at auto-encoding an MNIST image (0.539 accuracy in table 1)? It seems very low.
>
> Different tasks suit different model architectures. We would like to stress that the implementation of this experiment is based on the original MMVAE code and can be found in the attached source code.

---

### Official Review · AnonReviewer2 · 2020-10-28
**Good experimental defence of PoE, marginal originality**

**Rating:** 6
**Confidence:** 2

**Review:**

This paper proposes a variant of multimodal VAE model. It also argues in favour of product-of-experts approaches vs. additive mixture-of-experts ones.
The paper clearly frames the contribution within the relevant literature, the introduction is well written and the paper is well structured. Variants of multimodal VAE are also introduced (although acronyms are not made explicit, which would make the exposition clearer), and the new derivation is presented and explained.
Claims are supported by experimental results, that use MNIST, SVHN, CUB-Captions. A final discussion summarises the claimed contribution and advantages of the proposed approach.

Comments:
- Please make acronyms explicit early on to make the exposition clearer
- Figure 1 is a nice visualisation, very helpful
- Does the proposed approach generalise easily to more than two modalities?
- How well do you expect your approach to work on modalities presenting considerably different dimensionality (also in comparison with other approaches)?
- Quantitative results show comparable scorse between SVAE and VAEVAE variants. Where/why would SVAE be most beneficial/advantageous?
- Which applications would see MoE approaches outperform PoE? I think this point is important for completing the analysis and comparison of the two approaches. Is there a case where PoE and MoE perform comparably, and why would that happen?

---

> ### Author Response · Authors · 2020-11-24
> **Multimodal capacities; MoE vs PoE; modalities dimensions**
>
> We thank the reviewer for the helpful comments.
>
> >Please make acronyms explicit early on to make the exposition clearer
>
> We’ve added the acronyms explanations in the Background section of the manuscript.
>
> >Does the proposed approach generalise easily to more than two modalities?
>
> >Quantitative results show comparable scorse between SVAE and VAEVAE variants. Where/why would SVAE be most >
> beneficial/advantageous?
>
> Thank you for encouraging us to explore the multi-modal capabilities of SVAE and VAEVAE models. The manuscript is expanded with a section describing the 3-modalities architecture and the new MNIST-Split-3 experiment, please see the new section “Image and image: MNIST-Split” starting on page 5, Figure 5 and 6,  and Appendix D. Our SVAE architecture shows better performance in individual modalities reconstruction while having less parameters in the multi-modal case.
>
> >How well do you expect your approach to work on modalities presenting considerably different dimensionality (also in comparison with other approaches)?
>
> The two modalities in CUB-Captions dataset are very different in the modalities dimensionality (64643 for images vs 32 for text). One consequence of this is that the reconstruction errors of two modalities contribute unequally to the loss function. To fix this, the likelihood scaling coefficient for the image reconstruction error is introduced (32/64643 = 0.0026). Another example of such a dataset is images paired with classification labels, where the label is used as another modality of much smaller dimensionality. However if the dimensionality is reduced that much, the task becomes discriminative rather than generative. It was shown in the VAEVAE study that the multi-modal VAE performs worse than a classifier in that case.
>
> >Which applications would see MoE approaches outperform PoE? Is there a case where PoE and MoE perform comparably, and why would that happen?
>
> The case when both modalities carry complementary information is better captured by the PoE model since in order to make a confident prediction, both experts need to be in agreement. In case when the information is duplicated in two modalities, one confident expert is enough to make the correct prediction, and the MoE model is better suited. The two cases are exemplified by the MNIST-Split and MNIST-SVHN datasets. Not all the real-life datasets have a clear distinction, see also our reply to Reviewer 2. More complex datasets would often have both characteristics, making the model choice not obvious.

---

### Decision · Program_Chairs · 2021-01-07
**Final Decision**

**Decision:**

Reject

**Comment:**

The reviewers found this to be an interesting and clearly-written paper, but broadly agreed that it is not yet ready for acceptance. In particular, multiple reviewers felt that the experiments don't show clear benefits of the proposed SVAE approach when compared to the VAEVAE and other baselines; nor do they sufficiently back up the central claim regarding relative benefit of PoE vs MoE for either "AND" or "OR" relations. Hopefully the comments and suggestions from the reviewers, particularly regarding framing and experimental validation, will help in revising the paper.